# Cysteine Regulates Oxidative Stress and Glutathione-Related Antioxidative Capacity before and after Colorectal Tumor Resection

**DOI:** 10.3390/ijms23179581

**Published:** 2022-08-24

**Authors:** Feng-Fan Chiang, Te-Hsin Chao, Shih-Chien Huang, Chien-Hsiang Cheng, Yu-Yao Tseng, Yi-Chia Huang

**Affiliations:** 1Division of Colorectal Surgery, Department of Surgery, Taichung Veterans General Hospital, Taichung 40705, Taiwan; 2Department of Food and Nutrition, Providence University, Taichung 43301, Taiwan; 3Chiayi & Wanqiao Branch, Taichung Veterans General Hospital, Chiayi 60090, Taiwan; 4Department of Nutrition, Chung Shan Medical University, Taichung 40201, Taiwan; 5Department of Nutrition, Chung Shan Medical University Hospital, Taichung 40201, Taiwan; 6Department of Health Industry Technology Management, Chung Shan Medical University, Taichung 40201, Taiwan; 7Department of Respiratory Therapy, Taichung Veterans General Hospital, Taichung 40705, Taiwan

**Keywords:** cysteine, oxidative stress, antioxidant capacities, tumor resection, colorectal cancer

## Abstract

Cysteine might scavenge free radicals and is a limiting substrate for the cellular synthesis of glutathione (GSH). We investigated the association of cysteine with oxidative stress and GSH-related antioxidant capacity in colorectal cancer (CRC) patients. Plasma samples were drawn from 66 patients 1 day before (pre-resection) and 4 weeks after resection (post-resection). Tumor and adjacent normal tissues were collected. We measured levels of plasma and tissue cysteine, homocysteine, oxidative stress indicators (malondialdehyde, MDA; advanced oxidation protein products, AOPP), GSH, and antioxidant enzyme activities. After tumor resection, patients had significantly higher levels of plasma cysteine, homocysteine, MDA, AOPP, and GSH-related antioxidant enzyme activities when compared with pre-resection. Levels of cysteine, homocysteine, AOPP and all antioxidant capacity indicators in tumor tissue were significantly higher than those levels in the adjacent normal tissue. Plasma cysteine levels measured at pre-resection were positively associated with MDA levels in the tumor and in the adjacent normal tissues. Cysteine levels in tumor and adjacent normal tissues were significantly associated with tissue levels of homocysteine, almost as indicators of oxidative stress and antioxidant capacities. Cysteine in the circulation was likely utilized to mediate GSH-related antioxidant capacity and further cope with increased oxidative stress in tumor and adjacent normal tissues.

## 1. Introduction

Colorectal cancer (CRC) is one of the most common types of cancer, with rising mortality worldwide over the past decade [1]. Risk factors for CRC are obesity; male gender; high red meat intake, saturated fat and alcohol consumption; low fruit, vegetable, and fiber intake; prior diseases (i.e., diabetic mellitus, inflammatory bowel disease); smoking; and family history of CRC [2]. Greater oxidative stress or reduced antioxidant capacity leads to redox imbalance, resulting in tumor initiation and progression [3]. An imbalance between oxidative stress and antioxidant capacity has been suggested to cause CRC. The evidence is that healthy subjects have lower oxidative stress and higher antioxidant capacity in their plasma compared with CRC patients [4,5,6,7,8,9,10,11]. Tumor tissues also show higher oxidative stress and higher antioxidant capacity compared with adjacent normal tissues [6,12,13,14,15].

Adequate antioxidant defense can prevent threats of high oxidative stress to lower disease incidence or slow down disease progression. Regarding antioxidant defense mechanisms in the human body, glutathione peroxidase (GPx) and superoxide dismutase (SOD) situate at the first line, while glutathione (GSH, γ-L-glutamyl-L-cysteinylglycine) situates at the second line [16]. Under conditions of high oxidative stress, GSH offers electrons via GPx to reduce hydroperoxides, and then it self-oxidizes to glutathione disulfide (GSSG, an oxidized form of GSH). GSSG is further reduced back to GSH through catalyzation by glutathione reductase (GR). The thiol group (-SH) of cysteine is not only a major yet limiting substrate for the synthesis of GSH within cells, but it also directly scavenges free radicals; cysteine thiols thus play a crucial role in oxidative stress [17]. Studies indicated that CRC cells likely take up cysteine to protect themselves against high oxidative stress to maintain growth [18,19,20]. If tumor cells capture cysteine from circulation to protect themselves against high oxidative stress, theoretically, plasma cysteine levels in CRC patients might be associated with the risk of CRC. However, no significant association between plasma cysteine levels and CRC risk has been reported in previous studies [21,22,23,24] except one showing that high plasma cysteine levels significantly lower the risk of rectal tumors in postmenopausal women [25]. On the other hand, we observed in our previous study that CRC patients had higher levels of plasma cysteine and GSH compared with healthy subjects [22]. We could not rule out the possibility that cysteine mediates oxidative stress and antioxidant capacity through GSH-related defense systems to affect CRC risk indirectly.

The role that cysteine plays in regulating oxidative stress and antioxidant defenses in CRC patients has not been thoroughly investigated. Here, we aimed to investigate the association of cysteine with oxidative stress and GSH-related antioxidant capacity before and after tumor resection in CRC patients.

## 2. Results

Table 1 shows CRC patients’ demographic and clinical characteristics at pre- and post-resection times. Of the 66 CRC patients (34 men, 32 women), 32 had colon cancer and 34 had rectal cancer. More than half of the patients were diagnosed to be in CRC stages II and III. Patients during post-resection time had significantly lower body mass index (BMI), serum carcinoembryonic antigen (CEA), and carbohydrate antigen 19-9 (CA 19-9) after tumor resection compared with pre-resection.

Table 2 and Table 3 show the cysteine and homocysteine concentrations, oxidative stress indicators, and antioxidant capacity indicators in plasma, tumor, and adjacent normal tissues. Patients during post-resection not only had significantly higher plasma cysteine and homocysteine concentrations but also had significantly higher levels of malondialdehyde (MDA), advanced oxidation protein products (AOPP), trolox equivalent antioxidant capacity (TEAC), GSH, GSSG, GPx, GR, and glutathione *S*-transferase (GST) when compared with pre-resection. Their cysteine and homocysteine concentrations, levels of AOPP, and all antioxidant capacity indicators in tumor tissue were significantly higher than those in the adjacent normal tissue. The only exception was the significantly lower MDA levels in tumor tissue compared with the adjacent normal tissue.

Partial Spearman’s correlation was performed whether cysteine levels at pre-resection, tumor and adjacent normal tissues were associated with indicators of oxidative stress and antioxidant capacities (Table 4). More than half of our CRC patients were in stages II and III, and serum C-reactive protein (CRP) level slightly increased after tumor resection. We therefore considered the CRC stage and serum CRP level to be potential confounders for oxidative stress, and we adjusted for age, sex, smoking and drinking habits, cancer stage, and CRP levels when we performed the partial correlation analyses. Plasma cysteine level at pre-resection was positively associated with plasma pre-resection levels of homocysteine and TEAC levels, and it was also associated with plasma cysteine and homocysteine levels of post-resection. Plasma cysteine level was associated with homocysteine, GSH, and GSSG level at post-resection. Plasma cysteine level at pre-resection was not associated with cysteine and homocysteine levels in either tumor or adjacent normal tissues. Instead, it was positively associated with MDA, TEAC, and GSSG levels in tumor tissue and was further associated with MDA level in the adjacent normal tissue. Cysteine levels in tumor and adjacent normal tissues were significantly associated with tissue levels of homocysteine, which were likely indicators of oxidative stress and antioxidant capacity.

## 3. Discussion

CRC patients are well-known to have higher oxidative stress and lower antioxidative capacity compared with healthy subjects [4,5,6,7,8,9,10,11]. MDA, a byproduct of polyunsaturated fatty acid peroxidation, is considered a biological marker of oxidative stress [26]. AOPP are products of protein oxidation, likely reliable markers of oxidant-mediated protein damage [27]. To study oxidative stress and antioxidative capacity, we used levels of MDA and AOPP as oxidative stress indicators. Previous studies reported that plasma MDA and AOPP levels are elevated in CRC patients when compared with healthy controls [9]. These two plasma levels are simultaneously reduced at 24 h after tumor resection compared with pre-resection levels [11]. We wondered why our CRC patients had significantly higher plasma levels of MDA and AOPP after tumor resection compared with those before resection. It is worth noting that our CRC patients’ serum CRP and neutrophil-to-lymphocyte ratio (NRL) levels indicated that they were still in an inflammatory status after tumor resection. Ongoing inflammatory response might lead to elevated oxidative stress. Although surgery per se could induce oxidative stress [28], redox status is known to be regulated, returning to normal within 24 h after surgery [29]. Apart from surgery itself, inflammatory response, medication uses, or illness stress might all be possible causes of elevated oxidative stress in our CRC patients after tumor resection.

Regarding oxidative stress status in CRC tumor and in adjacent normal tissue, the colorectal tumor tissue has MDA levels higher than the adjacent normal tissue [13,14,15]. Veljković et al. [30] found that both MDA and AOPP are significantly higher in CRC tumor tissue than in normal tissue. In our present study, more surprisingly, such findings were opposite. Regarding tissue AOPP levels, we found oxidative stress status in tumor tissue was higher than in adjacent normal tissue. The level of MDA, other than AOPP, might be a potential non-invasive biomarker to differentiate tumor invasion depth or the presence of lymph node metastasis [9]. We thus speculated that one marker might be more sensitive than the other to reflect oxidative stress. Further study is warranted to confirm our speculation. Regardless of the status of oxidative stress in the tumor and adjacent normal tissues, colorectal tumor tissue had higher cysteine levels and antioxidant capacity compared with the adjacent normal tissue. Consistent with previous reports [6,13,14,15], malignant colorectal tissue likely utilizes cysteine and GSH-related antioxidant capacities to protect tumor against elevated oxidative stress.

Cysteine and GSH are essential antioxidants. Inadequate cysteine status may further disrupt cysteine and GSH metabolism to affect redox homeostasis. In our study, we did not recruit healthy controls. Total plasma cysteine concentration in healthy individuals is known to be 200 to 300 μmol/L [31]. Our CRC patients had mean plasma cysteine concentrations below 200 μmol/L, even though their mean levels were significantly higher, but still did not reach normal values after tumor resection. Our CRC patients’ deficient cysteine status might well be due to inadequate cysteine intake. As their dietary intake data were not recorded, this possibility was not assessed. In addition to dietary intake, cysteine is generated through the transsulfuration pathway. During transsulfuration, homocysteine transfers sulfur to form cysteine via cystathionine [32]. Approximately 50% of cysteine so generated can be utilized to synthesize GSH in hepatic cells [33]. Enzymes (e.g., cystathionine β-synthase and cystathionine γ-lyase) and coenzymes (e.g., vitamin B-6) are involved in the transsulfuration pathway [32]. Although we did not measure these enzyme activities and vitamin B-6 status in our study, cysteine concentrations were closely associated with homocysteine levels in both plasma and tissue. The generation of cysteine via the transsulfuration pathway was likely normal in our CRC patients. The transsulfuration pathway can be activated through enhanced de novo synthesis of cysteine during tumorigenesis, and cysteine deprivation in the transsulfuration pathway might be cytoprotective [20,34]. Thus, we propose that cysteine generation is likely enhanced through the transsulfuration pathway and that cysteine is used by tumor tissue to maintain its growth. This speculation is based on the changes of cysteine level in plasma between pre- and post-resection and the difference between tumor and adjacent normal tissues.

Yang et al. [35] observed that cysteine is likely more oxidized in CRC tissue than in non-tumor tissues, while cysteine is usually maintained in the reduced form in non-tumor tissue. If cysteinyl oxidation fosters free radical production and increases oxidative stress in tumor tissue, tumor cells can protect themselves against higher oxidative stress. We found that plasma cysteine levels at the pre-resection time were closely related to MDA levels in tumor and adjacent normal tissues, and cysteine levels were significantly associated with GSH-related antioxidant enzyme activities in tumor and adjacent normal tissues. Although we did not measure reactive oxygen species (ROS) levels in plasma and tissues, it is possible that plasma cysteine can move to tissue directly via its thiol group or that it indirectly mediates antioxidative capacity through a GSH-related defense system to cope with elevated oxidative stress in both tumor and adjacent normal tissues. The regulation of redox state by cysteine in the tumor and its adjacent normal tissue is likely complex during the tumorigenesis, and that needs to be further investigated.

The strength of our study was that tissue and plasma samples at the pre- and post-resection times were simultaneously collected. Study limitations existed. First, we did not collect healthy controls for reference. Second, we did not record dietary intake data. Third, the levels of ROS in plasma and tissues were not measured. Although we calculated sample size, 66 patients might not have adequate statistical power to reveal a more significant effect of cysteine on oxidative stress and GSH-related antioxidant capacities. Otherwise, the picture of cysteine status could be completely analyzed in our CRC patients.

## 4. Materials and Methods

### 4.1. Study Design and Sample Size Calculation

This was a cross-sectional study. To detect a significant correlation (*r* = 0.3) between the level of plasma cysteine and oxidative stress indicator (i.e., MDA), we needed 62 CRC patients to reach 80% statistical power (1 − β) and a two-sided α of less than 0.05.

### 4.2. Subjects

Consecutive patients were recruited from the division of colorectal surgery of Taichung Veterans General Hospital, Taiwan, with confirmed diagnosis of either colon or rectal cancer (International Classification of Diseases, Tenth Revision, Clinical Modification ICD-10-CM, codes C18–C20, respectively). All patients were scheduled to undergo tumor resection. Patients were excluded if they were younger than 20 years old or older than 80 years, pregnant or lactating, or had cardiovascular, liver, or renal diseases. Patients signed the informed consent form prior to enrolling the study. Our study was approved by the Institutional Review Board of Taichung Veterans General Hospital, Taichung, Taiwan (No. CF 19330A).

### 4.3. Data Collection and Biochemical Measurements

We recorded patients’ essential characteristics, such as age, sex, smoking and drinking habits, medications, and family history of colorectal cancer. Height, weight, systolic and diastolic blood pressures were measured one day before tumor resection (pre-resection) and 4 weeks after tumor resection (post-resection). BMI (kg/m^2^) was calculated as weight (kg) divided by height squared (m^2^).

Patients’ fasting blood samples were drawn and stored in vacutainer tubes (Becton Dickinson, Rutherford, NJ, USA) that contained appropriate anticoagulant or no anticoagulant as required at pre- and post-resection times. Serum levels of white blood cell, neutrophil count, and lymphocyte count, albumin, alanine aminotransferase, and creatinine were determined using an Automated Biochemical analyzer. We used the NRL as a reliable marker of ongoing cancer-related inflammation, and a valid indicator of prognosis of solid tumors, NRL ≥ 2.5–5.0, was adopted to serve as an independent prognostic index for CRC [36]. Therefore, this ratio was calculated using neutrophil and lymphocyte counts. Serum C-reactive protein was analyzed by particle-enhanced immunonephelometry using an Image analyzer, with a normal level at <0.3 mg/dL. Serum levels of CEA and CA 19-9, are widely used as tumor markers for CRC in clinical settings. We therefore determined these levels using an automated chemiluminescent immunoassay. The aforementioned analyses were carried out at our hospital’s department of pathology and laboratory medicine.

Colorectal tumor tissue and its adjacent normal tissue were obtained at the time of surgical resection. Patients’ diagnosis and cancer staging, histological type, and pathological grade of CRC were confirmed jointly by a pathologist and an oncologist. Tumor size and location were recorded accordingly. Resected tissues were homogenized on ice in 1 mL phosphate saline buffer (pH = 7.4). Homogenates were centrifuged at 13,000 rpm for 10 min at 4 °C. Supernatants were stored at −80 °C until analysis.

The following measurements were performed in the laboratory. Plasma and tissue cysteine levels and homocysteine levels were carried out under yellow lighting to minimize photodestruction and quantified by high performance liquid chromatography using fluorescence [37,38]. Plasma and tissue MDA and AOPP were determined to reflect oxidative stress. MDA was measured by the reaction of thiobarbituric acid at an excitation wavelength of 515 nm, and an emission wavelength of 555 nm using a fluorescence spectrophotometer [39]. AOPP were measured at a wavelength of 340 nm using commercial kits (Abcam company, Cambridge, UK). Plasma and tissue TEAC, GSH and GSSG levels, and activities of GPx, GR, GST, and SOD were used to reflect antioxidative capacity. TEAC was determined following the method of Arnao et al. [40]. GR activity was measured according to a published method [41]. GSH and GSSG concentrations (BioVision Incorporated, Milpitas, CA, USA), activities of GPx, GST, and SOD (Cayman Chemical Company, Ann Arbor, MI, USA) were analyzed using commercial kits.

### 4.4. Statistical Analyses

We used the SAS statistical software package (version 9.4, Statistical Analysis System Institute Inc., Cary, NC, USA). Shapiro–Wilk test was used to assess normality of distributions. Paired *t*-test or Wilcoxon signed-rank test was used to compare differences between pre- and post-resections. Categorical data were compared between groups using Chi-square or Fisher’s exact tests. The partial Spearman’s correlation was used to assess the associations between plasma or tissue cysteine concentrations and indicators of oxidative stress and also antioxidative capacity. These associations were analyzed after adjusting for age, sex, smoking and drinking habits, cancer stage, and CRP levels. Statistical significance was set at a two-sided *p* < 0.05.

## 5. Conclusions

CRC patients likely utilize cysteine in the circulation to mediate GSH-related antioxidative capacity and further cope with increased oxidative stress in tumor and adjacent normal tissues (Figure 1). Longer follow-up times (i.e., one or three years after tumor resection) would improve the monitoring clinical outcomes and the understanding of the roles of cysteine and GSH-related antioxidant capacity in the convalescence after tumor resection.

## Figures and Tables

**Figure 1 ijms-23-09581-f001:**
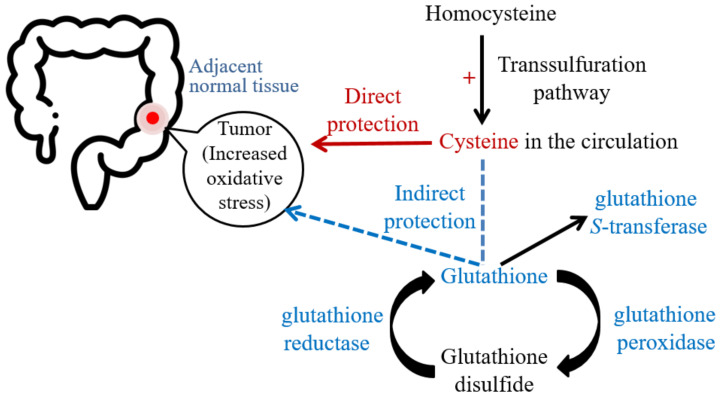
The possible effects of cysteine on oxidative stress and glutathione-related antioxidant capacities during the existence of colorectal cancer.

**Table 1 ijms-23-09581-t001:** Demographic and clinical characteristics of patients with colorectal cancer pre- and post-resection.

Parameters	Pre-Resection	Post-Resection
Age (y)	61.8 ± 11.1 (64.5)	
Male/Female	34/32	
BMI (kg/m^2^)	24.6 ± 4.6 (24)	23.3 ± 4.5 ^†^ (23.2)
Blood pressure (mmHg)		
systolic	133.8 ± 14.3 (135)	135.2 ± 17.8 (135)
diastolic	77.1 ± 10.9 (75)	79.4 ± 11.4 (79.5)
White blood cell (10^3^/μL)	7.6 ± 3.7 (6.8)	7.1 ± 2.4 (6.5)
Neutrophil-to-lymphocyte ratio	4.5 ± 4.4 (3.3)	3.5 ± 3.4 (2.5)
Serum C-reactive protein (mg/dL)	1.0 ± 1.8 (0.3)	1.2 ± 2.7 (0.3)
Serum albumin (g/dL)	4.1 ± 0.5 (4.1)	4.2 ± 0.4 ** (4.3)
Serum ALT (U/L)	21.8 ± 22.0 (16)	30.9 ± 36.8 * (20)
Serum creatinine (mg/dL)	0.9 ± 0.3 (0.8)	0.9 ± 0.3 (0.9)
Serum carcinoembryonic antigen (ng/mL)	56.8 ± 264.1 (4.7)	52.0 ± 353.3 ^†^ (2.3)
Serum carbohydrate antigen 19-9 (U/mL)	66.7 ± 294.5 (12.4)	36.3 ± 110.9 ** (10.3)
Family history of colorectal cancer (*n*, %)	11, 16.7%	
Stage at diagnosis (*n*, %)		
Stage 0	2, 3%	
Stage I	8, 12.1%	
Stage II	15, 22.7%	
Stage III	27, 40.9%	
Stage IV	14, 21.2%	
Cancer location		
Colon	32 (48.5%)	
Rectum	34 (51.5%)	
Smoking habit (*n*, %)		
Current	6, 9.1%	
Past	8, 12.1%	
No	52, 78.8%	
Drinking habit (*n*, %)		
Current	5, 7.6%	
Past	5, 7.6%	
No	56, 84.8%	

*n* = 66. Values are means ± standard deviation with median in the parentheses. BMI, body mass index; SBP, systolic blood pressure; DBP, diastolic blood pressure; ALT, alanine aminotransferase. * Values are significantly different from pre-resection; * *p* < 0.05; ** *p* < 0.01; ^†^
*p* < 0.001.

**Table 2 ijms-23-09581-t002:** Plasma cysteine, homocysteine, indicators of oxidative stress, and antioxidant capacities in patients with colorectal cancer.

	Pre-Resection	Post-Resection	*p* Value
Cysteine (μmol/L)	167.42 ± 35.07	186.09 ± 40.50	<0.001
Homocysteine (μmol/L)	9.94 ± 3.66	11.48 ± 3.88	<0.001
Oxidative stress indicators			
MDA (μmol/L)	0.97 ± 0.27	1.03 ± 0.30	0.044
AOPP (μmol/L)	385.06 ± 157.71	479.40 ± 244.00	0.012
Antioxidant capacities			
TEAC (μmol/L)	3909.91 ± 326.45	4486.69 ± 395.27	<0.001
GSH (μmol/L)	79.28 ± 45.11	114.69 ± 47.99	<0.001
GSSG (μmol/L)	482.33 ± 64.16	578.331 ± 76.43	<0.001
GSH/GSSG ratio	0.16 ± 0.07	0.20 ± 0.07	<0.001
GPx (nmol/mL/min)	166.74 ± 55.28	232.44 ± 66.93	<0.001
GR (nmol/mL/min)	55.96 ± 18.43	67.34 ± 29.07	<0.001
GST (nmol/mL/min)	26.01 ± 19.65	37.79 ± 21.62	<0.001
SOD (U/mL/min)	3.47 ± 1.76	3.61 ± 1.41	0.220

*n* = 66. Values are mean ± standard deviation. GSH, glutathione; GSSG, glutathione disulfide; MDA, malondialdehyde; AOPP, advanced oxidation protein products; TEAC, trolox equivalent antioxidant capacity; GPx, glutathione peroxidase; GR, glutathione reductase; GST, glutathione *S*-transferase; SOD, superoxide dismutase.

**Table 3 ijms-23-09581-t003:** Cysteine, homocysteine, indicators of oxidative stress, and antioxidant capacities in colorectal tumor tissues and adjacent normal tissues.

	Tumor Tissue	Adjacent Normal Tissue	*p* Value
Homocysteine (μmol/g protein)	1.00 ± 0.44	0.65 ± 0.28	<0.001
Cysteine (μmol/g protein)	35.03 ± 20.26	18.95 ± 10.01	<0.001
Oxidative stress indicators			
MDA (μmol/g protein)	0.20 ± 0.20	0.28 ± 0.51	0.017
AOPP (μmol/protein)	130.03 ± 42.34	86.42 ± 29.57	<0.001
Antioxidant capacities			
TEAC (μmol/g protein)	445.00 ± 177.28	394.59 ± 156.99	0.020
GSH (μmol/g protein)	10.90 ± 12.50	7.55 ± 4.09	0.001
GSSG (μmol/g protein)	187.96 ± 83.87	141.00 ± 42.97	<0.001
GSH/GSSG ratio	5.72 ± 4.08	5.33 ± 2.20	0.448
GPx (nmol/min/g protein)	136.01 ± 53.24	88.21 ± 35.71	<0.001
GR (nmol/min/g protein)	157.71 ± 76.98	134.20 ± 59.25	0.012
GST (nmol/min/g protein)	32.75 ± 11.44	28.49 ± 10.03	0.019
SOD (U/min/g protein)	17.10 ± 6.46	14.62 ± 4.15	0.003

Values are means ± standard deviation. GSH, glutathione; GSSG, glutathione disfluide; MDA, malondialdehyde; AOPP, advanced oxidation protein products; TEAC, trolox equivalent antioxidant capacity; GSH-Px, glutathione peroxidase; GSH-Rd, glutathione reductase; GSH-ST, glutathione *S*-transferase; SOD, superoxide dismutase.

**Table 4 ijms-23-09581-t004:** Partial Spearman’s correlations (*r_s_*) of cysteine level with indicators of oxidative stress and antioxidant capacities in patients with colorectal cancer at pre-resection, tumor tissues, and adjacent normal tissues after adjusting for potential confounders.

	Plasma Cysteine at Pre-Resection				Plasma Cysteine at Post-Resection	Cysteine in Tumor Tissue		Cysteine in Adjacent Normal Tissue	
	Plasma Level at Pre-Resection	Plasma Level at Post-Resection	Tumor Tissue	Adjacent Normal Tissue	Plasma Level at Post-Resection	Tumor Tissue	Adjacent Normal Tissue	Tumor Tissue	Adjacent Normal Tissue
Homocysteine	0.61 ^†^	0.36 **	0.18	0.07	0.39 **	0.74 ^†^	0.35 **	0.34 **	0.63 ^†^
Cysteine	−	0.45 ^†^	0.21	0.19	−	−	0.59 ^†^	0.59 ^†^	−
MDA	−0.23	−0.15	0.36 **	0.37 **	0.17	0.20	0.32 *	0.34 **	0.14
AOPP	0.24	−0.09	0.15	−0.01	0.03	0.05	0.32 *	−0.18	0.44 ^†^
TEAC	0.35 **	0.08	0.42 **	0.21	0.03	0.62 ^†^	0.42 **	0.48 ^†^	0.57 ^†^
GSH	0.00	0.00	0.29 *	0.09	0.31 *	−0.05	0.11	0.05	0.37 **
GSSG	0.04	−0.29 *	0.50 ^†^	0.23	0.34 **	0.56 ^†^	0.31 *	0.42 **	0.71^†^
GSH/GSSG ratio	−0.01	−0.06	−0.02	−0.14	0.25	−0.40 ^†^	−0.23	−0.19	−0.20
GPx	−0.07	0.09	0.24	0.15	0.25	0.48 ^†^	0.36 **	0.41 **	0.49 ^†^
GR	0.11	−0.08	−0.00	0.01	0.06	0.63 ^†^	0.51 ^†^	0.46 ^†^	0.80 ^†^
GST	0.22	−0.02	0.14	0.01	0.02	0.46 ^†^	0.33 *	0.23	0.52 ^†^
SOD	0.16	−0.03	0.26	0.07	−0.11	0.27 *	0.33 *	0.19	0.56 ^†^

*n* = 66. Values are partial Spearman’s correlation coefficients (*r_s_*). Adjusting for age, sex, C-reactive protein, smoking and drinking habits and cancer stage. MDA, malondialdehyde; AOPP, advanced oxidation protein products; TEAC, trolox equivalent antioxidant capacity; GSH, glutathione; GSSG, glutathione disulfide; GR, glutathione reductase; GPx, glutathione peroxidase. ** p* < 0.05; ** *p* < 0.01; ^†^
*p* < 0.001.

## Data Availability

The data are contained within the article.

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
