# Peer review of "Cysteine Regulates Oxidative Stress and Glutathione-Related Antioxidative Capacity before and after Colorectal Tumor Resection"

_ijms, 2022, doi:10.3390/ijms23179581_

Round 1

Reviewer 1 Report

This is very interesting manuscript where authors focus to see the scavenging properties of cysteine in response to free radical damage. Authors have opted several important parameters to signify the protective measures i.e. changes in the level of antioxidants, MDA, advanced oxidation protein products etc. Results obtained in this study shows that after tumor resection, patients significantly a significant level of changes measured in the patients for example, higher plasma cysteine and homocysteine levels, AOPP, and GSH-related antioxidant enzyme activities after compared with pre-resection. Overall, study shows scientific evidences to reach at some conclusion. I have some minor comments-

1. Abstract should be modified. Reduce the background and add few lines in methodology and conclusion. Structure the abstract considering background, methodology, results and conclusion (key findings).

2. In the result section of table 2, it would be useful to provide accurate P values (for example p= 0.001 or p< 0.001). Also, if possible indicate a significant level with different stars (for example- * p<0.01, **p< 0.001 etc.)

3. Authors claiming that Cysteine inhibits the free radical formation. Does authors measured reactive oxygen species (ROS) levels? Although, i can see several antioxidants parameters have been performed, even though, measuring ROS would be useful here.

4. Authors should a add a figure presenting a mechanism that how cysteine interacting with biological system? How it may block the free radical formation and how antioxidants (measured in this study) behaving against free radical formation.

Author Response

This is very interesting manuscript where authors focus to see the scavenging properties of cysteine in response to free radical damage. Authors have opted several important parameters to signify the protective measures i.e. changes in the level of antioxidants, MDA, advanced oxidation protein products etc. Results obtained in this study shows that after tumor resection, patients significantly a significant level of changes measured in the patients for example, higher plasma cysteine and homocysteine levels, AOPP, and GSH-related antioxidant enzyme activities after compared with pre-resection. Overall, study shows scientific evidences to reach at some conclusion. I have some minor comments-

1. Abstract should be modified. Reduce the background and add few lines in methodology and conclusion. Structure the abstract considering background, methodology, results and conclusion (key findings).

Ans: Thank you for the reviewer’s suggestion. Although we would like to add few lines in methodology and conclusion, we have been notified that the abstract should be a total of about 200 words maximum according to MDPI author's guidelines. Therefore, we have to reduce the content of abstract to be 200 words. Please see the revised abstract.

2. In the result section of table 2, it would be useful to provide accurate P values (for example p= 0.001 or p< 0.001). Also, if possible indicate a significant level with different stars (for example- * p<0.01, **p< 0.001 etc.)

Ans: Thank you for the reviewer’s suggestion. We provided the accurate p values in Table 2 and 3. In addition, we indicate a significant level with different stars in Table 1.

3. Authors claiming that cysteine inhibits the free radical formation. Do authors measure reactive oxygen species (ROS) levels? Although, I can see several antioxidants parameters have been performed, even though, measuring ROS would be useful here.

Ans: Unfortunately, we did not measure ROS levels in this study. Therefore, we add this as a limitation. Please see page 8.

4. Authors should add a figure presenting a mechanism that how cysteine interacting with biological system? How it may block the free radical formation and how antioxidants (measured in this study) behaving against free radical formation.

Ans: Thank you for the reviewer’s suggestion. Figure 1 shows the possible interaction of cysteine with oxidative stress and glutathione-related antioxidant capacity in plasma and tissues during the existence of colorectal cancer. Please see figure 1.

Reviewer 2 Report

The paper by Chiang et al. illustrates how cysteine regulates oxidative stress and glutathione-related anti-oxidative capacity before and after colorectal tumor resection. The work appears well written and well articulated, and is certainly worthy of publication in IJMS. I believe that it would have been even more complete if CRC patients had been compared to a healthy control group, in this respect the authors rightly place this point as a limitation of the study at the end of the Discussion.

Minor points

- According to MDPI author's guidelines the abstract should be a total of about 200 words maximum, please reduce it.

- The small number of CRC patients (66) should also be mentioned as a limitation at the end of the discussion. Large-scale studies confer a greater statistical value, certainly this study could represent a starting point for investigation with larger sample sizes.

Author Response

The paper by Chiang et al. illustrates how cysteine regulates oxidative stress and glutathione-related anti-oxidative capacity before and after colorectal tumor resection. The work appears well written and well articulated, and is certainly worthy of publication in IJMS. I believe that it would have been even more complete if CRC patients had been compared to a healthy control group, in this respect the authors rightly place this point as a limitation of the study at the end of the Discussion.

Minor points

1. According to MDPI author's guidelines the abstract should be a total of about 200 words maximum, please reduce it.

Ans: Thank you for the reviewer’s suggestion. The abstract has been reduced to 200 words. Please see the revised abstract.

2. The small number of CRC patients (66) should also be mentioned as a limitation at the end of the discussion. Large-scale studies confer a greater statistical value, certainly this study could represent a starting point for investigation with larger sample sizes.

Ans: Thank you for the reviewer’s comment. We add this point to be another study limitation. Please see page 8.
